# Family-Supportive Supervisor Behaviors and Psychological Distress: A Secondary Analysis across Four Occupational Populations

**DOI:** 10.3390/ijerph19137845

**Published:** 2022-06-26

**Authors:** Philip G. Bouleh, Shalene J. Allen, Leslie B. Hammer

**Affiliations:** 1Department of Psychology, Portland State University, Portland, OR 97207, USA; pbouleh@pdx.edu (P.G.B.); shallen@pdx.edu (S.J.A.); 2Oregon Institute of Occupational Health Sciences, Oregon Health & Science University, Portland, OR 97239, USA

**Keywords:** psychological distress, FSSB, occupational stress, mental health, social support

## Abstract

This study assessed the associations of employee’s perceptions of family-supportive supervisor behaviors (FSSB) and their psychological distress across four occupational populations (*n* = 3778): Information technology; healthcare; military-connected Veterans; and National Guard service members. Data were gathered and analyzed from four larger archival datasets to compare differences in these relationships. Results revealed significant negative relationships between employee reports of FSSB and their psychological distress within occupations, as expected. Furthermore, results revealed significant differences across occupational populations for employee reports of both FSSB and psychological distress. Hierarchical moderated multiple regression analyses were conducted to examine the extent of these mean differences across groups. Results revealed significant differences among these four groups such that the military-connected Veteran employees demonstrated significantly stronger associations of FSSB, and psychological distress compared to the other three occupations of information technology, healthcare, and National Guard service members. These findings suggest the importance of FSSB to worker psychological health across a variety of occupational populations, specifically noting the importance and presence of FSSB for Veteran employees’ psychological distress in civilian workplaces. Practical implications include the need for training leaders on how to better support employees’ work and non-work lives, mental health, and well-being.

## 1. Introduction

### 1.1. Psychological Distress and Work-Family Concerns

With both work-related challenges and environmental disasters, the psychological health and well-being of employees and their families is of growing concern in todays’ society [1]. In fact, many employees have recently faced significant impacts of distress as result of the COVID pandemic, ultimately affecting the way they are able to perform at work [2]. As such, there is a great need to understand how we can protect and support workers’ mental health and well-being [3], specifically, by examining factors that can improve their reports of psychological distress. For instance, mounting evidence suggests employees have experienced challenges managing their work and family responsibilities due to the growing incorporation of a 24/7 economy and the demands placed upon single parents and dual-earner couples who may be caring for children and/or elderly family members [4,5,6,7]. Such work-family conflict emerges when employees struggle to simultaneously fulfill their overlapping responsibilities between work and home life, and in turn problematically sacrifice the energy and time they invest in one domain for the other [8]. As a consequence, employees who experience greater levels of work–family conflict and decreased levels of organizational support are at greater risk of having increased negative affective experiences, such as anxiety and distress, as well as diminished organizational effectiveness [9,10]. For example, mothers who manage multiple roles composed of higher childcare responsibility in addition to professional work-related duties experience greater levels of psychological distress, depression and absenteeism, and lower levels of well-being [11,12]. This is especially relevant given that psychosocial occupational stressors (e.g., high work demands, long work hours, and low job control) have been consistently linked to poor psychological and physical health outcomes (e.g., cardiovascular disease, mortality, type 2 diabetes, depression) [13,14]. In consideration of the consequential relationship between work-family conflict and employee health and well-being, it is important to examine workplace conditions that can impact and alleviate the conflict between work–non-work domains and distress. For example, supervisor support for family is one factor that has been found to improve work–non-work lives of employees.

### 1.2. Family-Supportive Supervisor Behaviors (FSSB)

Social support has been extensively studied in the organizational science and occupational health literatures and is generally found to mitigate the bi-directional interference between work and family [15,16,17,18,19,20]. In particular, the provision of social support in the workplace via supervisors denotes a vehicle, which can provide important resources to employees [6]. FSSB is a more content-specific formulation of supervisor support that consists of four behavioral dimensions that can be implemented by supervisors to assist in balancing work-family demands [19,21]. The four dimensions comprising FSSB are emotional support, instrumental support, role modeling behaviors, and creative work-family management—all of which are characterized by a distinct set of leadership behaviors that improve the experience between the work and non-work lives of employees. For example, leaders express *emotional support* when they engage in authentic interpersonal interactions with their employees while creating a safe space to discuss and sympathize with their personal and family commitments. *Role modeling* behaviors are manifested as the exemplification of work–family balance in the daily life of a leader. *Instrumental support* consists of a manager proactively helping an employee manage work and family challenges, such as adjusting an employee’s schedule in response to their expressed work and family scheduling conflicts. Finally, *creative work-family management* is the proactive and innovative process, by which leadership collaborates with employees to restructure work for the purpose of maximizing the benefits experienced at work and at home [19,21]. Employee reports of FSSB have been correlated with important health, family, and workplace outcomes [22,23,24].

Training leaders on the importance of FSSB and in particular, specific behaviors to enact, has been shown to improve employee well-being outcomes [25,26]. For instance, research suggests that workplace contextual factors such as family-supportive organizational culture, work-family infrastructures, as well as individual-level factors (e.g., family life stage) contribute to FSSB [27]. This suggests that certain supervisor populations may be more or less likely to enact, and certain worker populations may be more or less responsive to, FSSB. Furthermore, certain occupations may be more or less conducive to FSSB. For example, information technology occupations are more conducive to work from home practices compared to traditional healthcare settings, and therefore, may have higher levels of FSSB enacted and received by supervisors and employees due to the nature of the work.

### 1.3. Current Study

In alignment with prior research [28], this study utilized a novel common measures methodological approach to investigate a repository of cross-sectional data from several large-scale studies for the purpose of examining the relationship between employee reports of FSSB and their reports of psychological distress among workers in various occupations. The goal of this study is to understand the relationship between supervisor support (i.e., FSSB) and psychological distress across diverse organizational contexts. Since FSSB is influenced by individual and contextual factors [27], this study provides insight into how different populations of workers perceive the presence of FSSB in their particular workplace, and how it relates to their psychological well-being.

Given FSSB principles are based in social support theory and literature [29], we anticipate that increased employee perceptions of FSSB will be related to decreases in their reports of psychological distress and improved employee psychological health and well-being. Social connections and connectedness are key levers in impacting psychological and physical health [30,31]. Therefore, it is important to understand not only the relationship between FSSB and psychological distress but discerning how this relationship may differ across occupational populations may point researchers and practitioners to workplace conditions that may be altered to facilitate higher levels of FSSB, which in turn, could help reduce psychological distress. As Straub [27] suggests, antecedents of FSSB include workplace contextual variables such as family-supportive organizational culture, top management openness, the organizational reward system, and access to work family infrastructure. While these contextual variables may explain differences in occupations that can impact levels of FSSB, when examining different occupational populations Straub [27] also suggests that individual factors such as life course stage, family life stage, social identification, and gender roles may also impact the enactment of FSSB. Thus, we aim to shed light on and explore how employee reports of FSSB and their psychological distress differs by occupational population. We examined the following hypothesis and research questions:

**Hypothesis** **1:**
*There will be a significant negative association between employee reports of FSSB and psychological distress across and within occupational populations.*


*Research Question 1a:* Is there a significant difference in employee reports of FSSB by occupational population?

*Research Question 1b:* Is there a significant difference in employee reports of psychological distress by occupational population?

*Research Question 2:* Does the relationship between employee reports of FSSB and psychological distress differ by occupational population?

As emphasized by Hanson and colleagues [28], the common measures methodology presents several empirical advantages given that it employs the same set of standardized measures both across and within populations. Additionally, population-based studies tend to be more expensive, may be limited to broader occupational categories, and have been lacking in their generalizability to U.S. populations given the majority have been conducted in Europe [28]. Therefore, the implementation of a common measures approach in the current study, enables a robust evaluation of the relationship between employee reports of FSSB and their psychological distress, specifically, examining data from four U.S. worker populations.

To the best of our knowledge, two prior studies utilized the common measures approach [28,32]. However, this is the first study to conduct a cross-sectional comparison utilizing baseline data to specifically examine the relationship between employee perceptions of FSSB and their psychological distress across four occupational samples. Our samples consist of military-connected and civilian samples, which lends to a unique assessment of the associations between perceptions of FSSB and psychological distress from National Guard service members, Veterans, healthcare, and information technology employees. This study additionally provides a benchmark for the reported rates of FSSB and psychological distress across such populations. In doing so, we provide further evidence for the generalizability of FSSB to enhance the psychological health and mental well-being of employees across multiple occupational roles, while also providing evidence for mental health prevention and intervention strategies for organizations, supervisors, and employees.

## 2. Materials and Methods

### 2.1. Overview of Study Populations

Demographic information across occupational populations revealed employees were primarily white (74.6%), female (53.7%), with an approximate age of 40 years old (*M* = 39.85, *SD* = 11.10). Below we briefly describe each study population (also see Table 1 for study and population overview). See Table 2 for a complete overview of sample characteristics and descriptive information within and across occupational populations.

***Study for Employment Retention of Veterans (SERVe).*** SERVe was a randomized control trial (RCT) designed to evaluate the impact of a Veteran Supportive Supervisor Training on the health and well-being of Veteran employees in civilian organizations. The sample consisted of 512 baseline participants who were post-911 Veterans, employed across 35 organizations in the Pacific Northwest, indicating they were primarily white (80.6%), males (83.6%), with an approximate age of 39 years old (*M* = 39.00, *SD* = 9.39). See Table 2 for demographic information. For a detailed explanation of study methodology see [33]. 

***Military Employee Sleep and Health Study (MESH).*** MESH was a cluster RCT designed to improve the sleep, health and well-being of 919 baseline participants who were full-time military service members of the National Guard. Service Members indicated they were primarily white (82.0%), males (75.6%), with an approximate age of 37 years old (*M* = 37.39, *SD* = 8.92). See Table 2 for demographics. For a detailed description of study methods see Hammer and colleagues [22]. 

***Work, Family, and Health Network Study (WFHS).*** WFHS is a National Institutes of Health (NIH, Kansas City, MO, USA) and Center for Disease Control and Prevention (CDC, Atlanta, GA, USA) funded study evaluating a work-family support intervention designed to improve the health and well-being of managers, employees, and their families. The study consisted of two baseline employee samples, namely 823 information technology (IT) employees working at a Fortune 500 firm, and 1524 healthcare employees working in long-term care positions within nursing homes. See Bray and colleagues [34] for an in-depth description of study methods.

***Information Technology Sample***. Of the 823 IT employee sample collected from the WFHS, participants reported they were primarily white (71.2%), males (60.9%), with an approximate age of 46 years old (*M* = 45.66, *SD* = 9.20). See Table 2 for a breakdown of demographic and population characteristic information. 

***Healthcare Sample***. Of the 1524 healthcare employees working in long-term care positions who participated in the WFHS, participants primarily indicated they were white (70.1%), females (91.8%), with an approximate age of 38 years old (*M* = 38.46, *SD* = 12.58). See Table 2 for detailed characteristic and demographic information.

### 2.2. Common Measures across Study Populations

***Family-Supportive Supervisor Behaviors (FSSB).*** Employees rated the extent to which they agreed that their direct supervisor exhibited family-supportive supervisor behaviors with the following four items on a 5-point scale (1 = *strongly disagree*, 5 = *strongly agree*): “Your supervisor makes you feel comfortable talking to him/her about your conflicts between work and non-work”, “Your supervisor demonstrates effective behaviors in how to juggle work and non-work issues”, “Your supervisor works effectively with employees to creatively solve conflicts between work and non-work”, and “Your supervisor organizes the work in your department or unit to jointly benefit employees and the company” [21]. See Table 3 for alpha coefficients.

***Psychological Distress scale-K6.*** Employees rated how often they experienced non-specific psychological distress in the last month with the following six items on a 5-point scale (1 = *None of the time*, 5 = *All of the time*): “Feel so depressed that nothing could cheer you up?”, “Feel hopeless?”, “Feel restless or fidgety?”, “Feel that everything was an effort?”, “Feel worthless?”, “Feel nervous?” [35]. See Table 3 for alpha coefficients.

### 2.3. Analyses

Means, standard deviations, correlations, and reliabilities for study variables were computed to create individual profiles for study samples across four populations (IT, healthcare, Veteran, service member). Initial reliabilities were computed using Cronbach’s alpha across integrated measures. Pearson correlations were computed to determine if there was a significant association between employee perceptions of their supervisors’ FSSB and employee self-reported psychological distress across occupational populations. A one-way analysis of variance (ANOVA) was conducted to determine the differences between means of the four occupational groups for FSSB and psychological distress measures. Additionally, to assess differences of the association of employee perceptions of FSSB and psychological distress between occupational groups, hierarchical moderated multiple regression analysis was implemented. 

Data were analyzed in SPSS Version 27 for descriptives, correlations, reliabilities and for ANOVA analyses. Furthermore, data were analyzed in Mplus Version 8 [35] to assess the moderated regression models. Alpha significance was set at *p* = 0.05 for two-tailed tests for determining statistical significance. Descriptive information on demographic and work characteristics across the four study samples (*n* = 3778) is found in Table 2.

### 2.4. Measurement Integration

Baseline item responses across studies (WFHS, SERVe, and MESH) were combined to create composite scale scores for the 4-item FSSB-SF measure [21] and the 6-item psychological distress measure [35]. To standardize item and scale integration across populations, data discrepancies were corrected before running analyses. For example, the FSSB measure, retrieved from the WFHN study, originally sampled participants on a 5-item scale, thus, composite scores for the integrated data excluded item “Respondent and supervisor talk effectively to solve conflicts”. After the FSSB scale was re-scaled to the 4-item measure, items were reversed coded and composite scores were obtained to standardize measures across groups. To integrate data for the K6 psychological distress measure item responses from the WFHN study were reverse coded for all 6-items before obtaining composite scores. In addition, item responses were originally scaled from 1 to 4 for participant data on the K6 measure from SERVe. Thus, for consistency, all 4 item response options were inflated to standardize composite scores. New scale scores were then computed for SERVe participant responses on the K6 measure prior to integration. The measures, including item responses, composites, and response options for FSSB and psychological distress were then integrated into one composite measure of FSSB and psychological distress across all study samples.

## 3. Results

### 3.1. Association of FSSB and Psychological Distress 

A Pearson correlation coefficient was computed to examine the relationship between employee reports of FSSB, *M* = 3.84, *SD* = 0.90 and their self-reports of psychological distress, *M* = 1.87, *SD* = 0.73. Results indicated that there was a significant negative correlation between employee perceptions of FSSB and their psychological distress, *r* = −0.20, *p* < 0.01, 95% CI [−0.23, −0.17] across all occupational populations. Approximately 4% of the variance was shared between FSSB and psychological distress in the combined sample, *r*^2^ = 0.04. Pearson correlation coefficients obtained similar findings within occupational groups: IT (*r* = −0.12, *p* < 0.01, 95% CI [−0.19, −0.05]), healthcare workers (*r* = −0.13, *p* < 0.01, 95% CI [−0.18, −0.08]), Veteran employees (*r* = −0.21, *p* < 0.01, 95% CI [−0.29, −0.13]), and service member employees (*r* = −0.18, *p* < 0.01, 95% CI [−0.24, −0.12]). Thus, Hypothesis 1 was confirmed; there was a significant negative association between FSSB and psychological distress across and within occupational populations.

### 3.2. Mean Difference Comparisons

A one-way analysis of variance tested whether employees from the four occupational groups (IT, healthcare, Veteran, service member) differed significantly on their self-reported perceptions of FSSB (RQ2a). Service members reported higher FSSB (*M* = 4.10, *SD* = 0.94), compared to IT employees (*M* = 3.83, *SD* = 0.82), Veterans (*M* = 3.81, *SD* = 0.93), and employees in healthcare (*M* = 3.69, *SD* = 0.88). Results indicated that these means differed significantly, *F*(3,3748) = 41.54, *p* < 0.001. The proportion of variance in FSSB accounted for by type of occupational group was approximately 3%, *η*^2^ = 0.03. The Tukey post hoc tests indicated that reports of FSSB across service member employees and healthcare employees differed significantly from Veterans and information technology employees (*p* < 0.05).

Similarly, a one-way analysis of variance was used to test whether employees from the four occupational groups differed significantly on their self-reported psychological distress levels (RQ2b). Veteran employees reported higher psychological distress (*M* = 2.11, *SD* = 1.00), compared to healthcare employees (*M* = 1.98, *SD* = 0.72), employees in IT (*M* = 1.81, *SD* = 0.54), and service member employees (*M* = 1.62, *SD* = 0.65). Results indicated that these means differed significantly, *F*(3,3756) = 70.66, *p* < 0.001. The proportion of variance in psychological distress accounted for by type of occupational group was approximately 5%, *η*^2^ = 0.05. The Tukey post hoc tests indicated that reports of psychological distress across all four occupational categories differed significantly from each of the groups (*p* < 0.05).

### 3.3. Hierarchical Moderated Multiple Regression 

Finally, to determine if the relationship between employee perceptions of FSSB and psychological distress differed across employee occupational populations (RQ3), hierarchical moderated regression analyses were conducted to examine if any one or more of these group memberships impacted the strength of the relationship between these two variables. Psychological distress was modeled as the outcome and FSSB was included as the predictor for the model. The moderator was dummy coded into four dummy variables: P1 = IT; P2 = healthcare; P3 = Veteran; P4 = service member; representing four categories with the Veteran sample serving as the reference group.

Interaction terms were added into the second model and effects were constrained across the groups to assess and compare the full vs. reduced model results for the reference group (i.e., Veteran sample) compared to the other three occupational categories. Results of the categorical moderation revealed the relationship between FSSB and psychological distress was significantly different for the Veteran sample compared to the other three occupational groups, as indicated by the significant Δ*R*^2^ = 0.002, (Δχ^2^ = 9.04, *df* = 3, *p* = 0.029), such that the negative relationship between employee reports of FSSB and psychological distress for those employees in the Veteran sample was significantly stronger compared to the other three samples. Results suggest Veteran employees reporting low FSSB from their civilian supervisors have significantly stronger reports of psychological distress compared to the other three samples, lending support to the importance of supervisors’ support for work–non-work lives of Veteran employees especially. Results of this analysis appear in Table 4. Figure 1 shows the relationship of these results across the association of FSSB and psychological distress for the Veteran sample compared to the other three groups. Additionally, when excluding the Veteran sample from model comparisons, there were no significant differences in the strength of the relationship between FSSB and psychological distress across the other three occupational populations samples.

## 4. Discussion

This study used a common measures methodological approach to explore the relationship between employee perceptions of FSSB and their psychological distress both within and across four occupations. Our findings revealed a consistently significant, negative relationship between employee perceptions of FSSB and their psychological distress within and between samples of healthcare, IT, service member and Veteran employees. These findings provide evidence for the importance of FSSB in relation to employee psychological distress across diverse occupational roles and environments. Furthermore, mean comparison findings revealed healthcare workers’ perceptions of FSSB were lower when compared with IT, Veteran, and service member populations, consequently signaling the need to intervene in this occupation distinctly, consistent with findings on high employee burnout and psychological distress among healthcare workers during the coronavirus pandemic (COVID-19) [36,37]. Additionally, results highlight the unique needs of supervisors in civilian organizations offering support to Veteran employees, as demonstrated by the findings that Veteran employees who reported lower FSSB from their supervisors also reported higher psychological distress compared to the other three samples. This research has important implications for future occupational health research and leader-centered workplace interventions.

Consistent with previous findings, this study reasserts the negative association of FSSB with markers of poor mental health in employees [25,38,39], while simultaneously calling for further investigation of the moderators (e.g., occupational context, population characteristics) of the psychological benefits of workplace leadership support. For example, FSSB has been previously shown to increase psychological well-being in healthcare employees [39] and diminish psychological distress in IT employees working less hours and under low-strain conditions [40]. Furthermore, FSSB was found to be negatively associated with psychological distress in an active-duty military population [41] and supervisor support has been shown to reduce negative emotions in Veterans screening positive for post-traumatic stress disorder (PTSD) [42]. High-risk military populations, such as Veterans and active-duty military personnel, typically report more adverse mental health outcomes when compared to their civilian counterparts [43] and may therefore require more leadership support that is sympathetic to their work and non-work needs in civilian contexts. Thus, this may in part explain why the relationship between FSSB and psychological distress was strongest among Veterans in this study, with the addition of the military’s influence on Veterans’ strong predisposition towards leadership structures [44]. However, Veterans also reported the highest rates of psychological distress compared to the other three groups—and while prior evidence has shown Veterans experience immense benefits across work, and the physical and psychosocial outcomes from having more supportive leaders in the workplace [45,46], the nature and severity of the psychological distress experienced by Veteran employees may require additional intervention [47]. For example, combat-exposed Veterans report relatively high rates of PTSD [48], a high comorbid disorder with depression [49] that is known to diminish the beneficial effects of social support on psychological distress [50].

It is also important to note the service member sample did not demonstrate comparable rates of psychological distress to Veterans even while reporting the highest FSSB levels. The lower rates of psychological distress in the service member sample may be best accounted for by differences in combat or trauma exposure, branch of service, and history of deployment [51,52] while their higher FSSB scores may be associated with their ongoing engagement in reservist activities in contrast from military separation that is experienced by Veterans.

As suggested by Straub [27], our study reveals the degree of benefit afforded by FSSB for psychological health. For example, a rigorous RCT demonstrated a larger psychological benefit of FSSB-centered training interventions for employees with increased elder and childcare responsibilities off the job [25]. Furthermore, another study demonstrated that an IT setting characterized by lower job demands and higher job control contributed to greater declines in employee psychological distress over 18 months when compared to a healthcare setting with higher job demands and lower control [40]. This suggests the differences in reported rates of FSSB and their relationship with improved psychological health may in part be impacted by occupational factors such as the structure/nature of work, as demonstrated in this study. For instance, our healthcare sample, which is predominantly female (91.8%), reported the lowest rate of FSSB (*M* = 3.69, *SD* = 0.88) in contrast with the predominantly male sample (75.6%) of military service members who reported the highest levels of FSSB (*M* = 4.10, *SD* = 0.94). In addition, the healthcare sample had the second highest scores of psychological distress, coming only second to Veterans. As a possible explanation for the health care sample’s lowest reports of FSSBs—gender role ideologies may predispose women to greater household and family caretaking responsibilities [27], which in turn create a greater need for FSSB and the more sensitized perception of the lack of FSSB in a predominantly female-led work context. Furthermore, the healthcare sample reported the lowest rate of marriage or cohabitation while having the second highest average number of children, suggesting the increased likelihood of being a single caretaker. Consequently, FSSB may be expressly critical for psychological well-being within occupational populations that are female-led and composed of single parent households, especially in workplaces characterized by high-strain (i.e., long hours and low control), such as those seen in healthcare settings [40].

### 4.1. Implications

A growing body of evidence derived from rigorous RCTs suggests that training supervisors in workplace support interventions (e.g., FSSB training’s) provides an effective means for supporting employee psychological health and well-being outcomes [22,24,33,34,46,53,54]. This research suggests the important role that the workplace can play in improving the mental health of employee populations, and highlights the potential of FSSB training interventions for ameliorating the negative effects of psychological distress. Furthermore, this research suggests that not only do employee reports of FSSB and psychological distress differ by occupational population, but that the relationship between FSSB and psychological distress differs by occupation. This is not surprising given the extensive research on the psychosocial stressors such as demands, control, and social support at work as significantly impacting psychological health and well-being of workers [14]. Thus, our findings suggest that continued work at mitigating the impact of such psychosocial stressors is an important strategy, regardless of the occupational population. 

In addition, employees may benefit from supervisors who enact FSSB as the conditions of organizations and the workforce environment may have more recent challenges brought about by COVID-19. Furthermore, there has been an increase in the number of workers working from home (WFH), with an estimated 42% of the U.S. workforce being driven to adapt to the new conditions associated with full-time WFH [55]. For example, during the peak of the pandemic, schools and daycares closed doors, childcare increased [56,57] and parents were faced with having to manage greater responsibilities within the home environment, creating conflicts between competing work and family demands [57,58]. And while WFH is related to increased control over work and positive outcomes for many, a growing percentage of parents reported deteriorating mental health for themselves and worsening behavioral health for their children [59]. Among the various known risks of WFH stated above, are those psychosocial in nature (e.g., social isolation, work-home lines blurred) [60], which can exasperate work-family demands [61], thereby requiring greater work-home boundary management support from employers [62]. On the other hand, frontline workers (e.g., healthcare employees) have also confronted their own set of work-related mental health challenges [37], while under the pressures of increased risk of infection [63] and fears of transmitting COVID-19 to family members [64]. As such, leaders in organizations across the U.S. need innovative solutions and resources for maneuvering through the urgent workplace demands presented by COVID-19 while mitigating risks to the well-being of employees and their families [62]. Providing more leadership support around such work-life challenges has been called for by the popular press and has been shown to be an effective way of reducing distress and improving well-being among workers [23,25]. It is well-established that social isolation and decreased social connectedness are related to increased psychological distress [65]. In fact, Holt-Lunstad [65] called for prioritization of social factors when addressing public health during, and in recovery from, the pandemic. Such a call is consistent with the focus of this study, which examines work as one of the missing links in improving mental health [66]. Furthermore, supervisors should be trained to look for signs of increased psychological distress during times of global crisis, and proactively destigmatize the self-disclosure of higher rates of anxiety or depression by their employees. This can in turn create a more transparent and authentic work environment where signs of reduced mental health are not viewed as a threat to employment, but as an opportunity to build trust with supervisors who can intervene by encouraging access to mental health resources.

### 4.2. Limitations

Limitations of this study include being focused on only four specific occupational populations and therefore, generalizing to other occupations may be limited. However, we did analyze occupations that were conceptually different from one another (e.g., desk jobs in information technology compared to high-risk occupations, such as the military) to enhance generalizability of findings. Furthermore, common method bias is a limitation given combined data were based on baseline cross-sectional samples. However, evidence of differences across populations suggests that this is not a critical limitation. Finally, we do have a methodological limitation relevant to reverse scoring items for analysis. For instance, for validation and standardization purposes across transformations, items were reversed coded for both FSSB and the K6 psychological distress scale, as such, a possible consequence contributing to the reduction in reliability. Lastly, it is important to note that the data analyzed in this study were gathered during the course of four separate studies occurring at different time points. Therefore, the differentiation in timing and research methodologies between studies should be considered when interpreting this study’s results.

### 4.3. Future Directions

Future leadership support interventions would benefit from a greater understanding of population-dependent contextual and individual factors that encourage supervisors to engage in FSSB with respect to specific employee characteristics and needs. How does one intervention fair in one population versus another, and what accounts for its differential effects on psychological well-being? For example, Kossek [25] found an “organizational job resource-enhancing intervention” that is predicated on FSSB was more effective for diminishing perceived stress than psychological distress, and suggested introducing an intervention that more precisely targets symptoms of psychological distress relevant for individuals caring for children and elders. Similarly, Veterans with high levels of psychological distress may benefit from interventions that encourage supervisors to recognize and ask Veterans about their mental health over and above focusing on work and family. Equally so, more research is needed on the effects of supervisor interventions for supervisors themselves. For example, one study demonstrated that a supervisor intervention may be detrimental to a supervisor’s work-to-family conflict and organizational commitment while simultaneously diminishing employee psychological distress [67], yet this has not been investigated extensively. As such, further research on the specific cultural and contextual factors of occupations such as healthcare, military service members, Veteran, and IT employee populations may shed light on the unique attributes that influence FSSB in their respective settings. Future research should also investigate how specific employee mental health symptoms (e.g., depression, PTSD, anxiety symptoms) differentially moderate the effectiveness of leadership support in various populations, and whether additional clinical interventions may help maximize the benefits of supervisor support in similar higher-risk employee populations.

## 5. Conclusions

This study provides evidence for the importance of FSSB in relation to psychological distress within and across multiple occupational populations, especially for high-risk populations such as Veteran employees working in a civilian context. Furthermore, this study presents a benchmark for the association between FSSB and psychological distress in a large, combined sample of IT, healthcare, Veteran, and service member employees (*n* = 3778). Our findings suggest the importance for training supervisors on how to enact FSSB, which may have an impact on lowering psychological distress and improving employee mental health in various occupational contexts. Most importantly, our results draw attention to the indispensable nuances of individual and occupational characteristics that may affect the relationship between supervisor support for family and employee psychological distress. This study has implications for helping shape future workplace supervisor support research and interventions and the importance of taking workplace context into consideration in the developments of such interventions. Our study suggests that workplace efforts to train supervisors on how to be more supportive of work–non-work lives of employees with an emphasis on support for the psychological health and well-being of employees may be a critical strategy for improving mental health and well-being of workers going forward.

## Figures and Tables

**Figure 1 ijerph-19-07845-f001:**
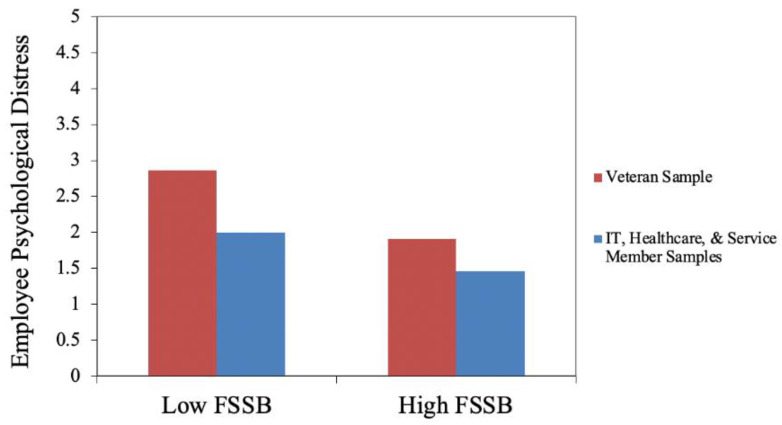
Model Comparisons of FSSB and Employee Psychological Distress of the Veteran sample compared to the other three occupations (IT, Healthcare, Service Member populations), graphically revealing the negative relationship between employee reports of FSSB and psychological distress for Veteran employees compared to the other three occupations (IT, Healthcare, Service Member).

**Table 1 ijerph-19-07845-t001:** Study Breakdown and Population Overview.

Reference Number	Citation	Study Name	Study Design	Sample(s)	Method
[33]	Hammer et al., 2017	Study for Employment Retention of Veterans (SERVe)	Randomized controlled trial	Post-9/11 veterans working in civilian organizations working at least 20 h per week (*n* = 512)	A Veteran Supervisor Support Training intervention was implemented to supervisors across 42 civilian organizations to determine overall effectiveness for improving the lives of veteran employees with data collected at baseline, 3-month, and 9-months.
[22]	Hammer et al., 2021	Military Employee Sleep & Health Study (MESH)	Cluster randomized controlled trial (cRCT)	Full-time National Guard service members from Army and Air branches (*n* = 919)	Large-scale Intervention study with training content focused on FSSB and Supervisor Support for sleep health in tandem with sleep tracking and individualized sleep health feedback for employees and supervisors, with data collected at baseline, 4-month, and 9-months.
[34]	Bray et al., 2013	Work, Family, and Health Network Study (WFHS)	Group-randomized field experiment	Low and high-wage employees across two large companies from information technology (*n* = 823) and healthcare care (*n* = 1524) working 22 h or more per week.	Intensive intervention study aimed at enhancing employees’ control over their work time by implementing participatory work redesign activities with training supervisors on support for employee’s family lives, with data collected at baseline, 6-, 12-, and 18-months.

**Table 2 ijerph-19-07845-t002:** Means, Standard Deviations, and Percentages for Employee Study Demographic and Work. Characteristics by Occupational Group.

Variable	Combined Sample (*n* = 3778) *M* (*SD*) %	Information Technology (*n* = 823) *M* (*SD*) %	Healthcare (*n* = 1524) *M* (*SD*) %	Veteran (*n* = 512) *M* (*SD*) %	Service Member (*n* = 919) *M* (*SD*) %
Female	53.7%	39.1%	91.8%	16.4%	24.2%
Male	46.3%	60.9%	8.2%	83.6%	75.6%
Age	39.85 (11.10)	45.66 (9.20)	38.46 (12.58)	39.00 (9.39)	37.39 (8.92)
Race/Ethnicity					
White	74.6%	71.2%	70.1%	80.6%	82.0%
Latinx or Hispanic	8.9%	6.6%	13.4%	1.0%	8.0%
Black or African American	6.9%	2.9%	13.9%	1.8%	1.4%
Asian or Asian Indian	6.6%	20.9%	3.4%	2.0%	1.9%
American Indian or Alaska Native	0.7%	0.1%	0.6%	1.6%	0.8%
Native Hawaiian or Pacific Islander	1.0%	1.7%	0.7%	0.2%	1.2%
More Than One Race or Other Race	8.0%	3.0%	11.2%	12.9%	4.7%
Married or Cohabitating	72.4%	79.3%	62.9%	77.7%	79.0%
Number of children	1.09 (1.20)	0.99 (1.08)	1.03 (1.18)	0.94 (1.13)	1.33 (1.30)
Some college, technical school, or degree	80.5%	96.2%	61.7%	94.5%	89.9%
Hours worked per week	40.90 (7.28)	45.42 (5.69)	36.88 (7.29)	42.36 (6.54)	42.72 (5.23)
Years at current job or tenure	7.55 (7.80)	13.34 (9.22)	5.93 (6.53)	8.05 (5.97)	4.88 (5.75)

*Note:* Data represents Baseline information.

**Table 3 ijerph-19-07845-t003:** Means, Standard Deviations, and Reliabilities for FSSB and Psychological. Distress Measures.

Employee Group	FSSB				Psychological Distress			
	*M*	*SD*	*N*	*α*	*M*	*SD*	*N*	*α*
Information Technology	3.83	1.32	821	0.88	1.81	0.54	823	0.77
Healthcare	3.65	1.10	1515	0.89	1.98	0.72	1522	0.83
Veteran	3.81	0.94	508	0.93	2.11	0.75	509	0.90
Service Member	4.10	0.94	916	0.95	1.62	0.65	900	0.87
Combined Sample	3.84	0.90	3752	0.91	1.87	0.73	3757	0.85

*Note:* Data represents Baseline information.

**Table 4 ijerph-19-07845-t004:** Multiple moderated regression results.

Variable	*B*	*SE*	*β*	*R* ^2^	Δ*R*^2^
*Model 1*				0.081 **	
IT	−0.30 **	0.05	−0.17 **		
Healthcare	−0.15 **	0.05	−0.10 **		
SM	−0.45 **	0.05	−0.27 **		
FSSB	−0.13 **	0.01	−0.17 **		
Constant	2.11 **	0.04	2.88 **		
*Model 2*				0.083 **	0.002 *
IT	−0.70 **	0.20	−0.40 **		
Healthcare	−0.55 **	0.20	−0.37 **		
SM	−0.86 **	0.20	−0.50 **		
FSSB	−0.22 **	0.05	−0.28 **		
IT X FSSB	0.11 *	0.05	0.24 *		
Healthcare X FSSB	0.11 *	0.05	0.27 *		
SM X FSSB	0.11 *	0.05	0.26 *		
Constant	2.11 **	0.04	2.88 **		
*Model 3*				0.076 **	0.000
IT	−0.20	0.13	−0.11		
SM	−0.40 **	0.14	−0.23		
FSSB	−0.14 **	0.02	−0.17		
IT X FSSB	0.00	0.03	0.01		
SM X FSSB	0.01	0.03	0.03		
Constant	2.00 **	0.02	2.73 **		

*Note:* IT = Information Technology, SM = Service Member sample. FSSB was centered. For Models 1 and 2 the veteran sample was modeled as the reference category. The Healthcare group was modeled as the reference category in Model 3. * *p* < 0.05. ** *p* < 0.01. *Ns* range = 3752–3778.

## Data Availability

Data available upon request. Permission was available for re-analysis of data across studies. Data use security protocols were signed before permission was granted.

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
