# Peer review of "Family-Supportive Supervisor Behaviors and Psychological Distress: A Secondary Analysis across Four Occupational Populations"

_ijerph, 2022, doi:10.3390/ijerph19137845_

Round 1

Reviewer 1 Report

The authors have presented a very good manuscript with detailed methodology and presentation of findings. The work adds to the already available body of knowledge. The authors should consider the few comments below;

1. Consider exploring the effect of different time points of collection of the different time points. I note that given the different time points, there might have been interventions put in place, hence there might have been changes across time for the different surveys. If not possible to do this then it should be listed as a limitation.

2. Did the authors consider the impact of reverse coding?. If not it might be good to explore or at least mention it as a limitation.

3. In my considered view I don't think  scale inflation should be a limitation since the authors utilized known and tested quantative methodology. 

4. The authors should further explore the potential implication of the interesting finding "when excluding the Veteran sample from model comparisons, there were no significant differences in the strength of the rela tionship between FSSB and psychological distress for the other three occupational populations samples" in the discussion section.

Reviewer 2 Report

This manuscript addresses an important topic of employees’ perception of FSSB and their psychological concerns across four different occupational populations. The authors have described the background information systematically with appropriate references. The hypothesis is also well stated. The materials and methods section is also meticulously written. The authors have described the results in an easy-to-understand manner using suitable statistics. The discussion states different reasoning and combines similar studies to understand the current work properly. The authors appreciate their limitations, have addressed them accurately, and have also given a plan for their future directions. Overall this is a unique and robust manuscript. 

Reviewer 3 Report

The Relationship Between Family-Supportive Supervisor Behaviors and Psychological Distress Across Four Occupational Populations

I thank you for the opportunity to comment this article.

Authors conclusion

“This study provides evidence for the importance of FSSB in helping mitigate psychological distress within and across multiple occupational samples, especially for high-risk populations such as Veteran employees working in a civilian context. Furthermore, this study presents a benchmark for baseline rates of FSSB and psychological distress in a large 494 sample of IT, healthcare, Veteran, and service member employees.” 

Comments

Although the subject of the research is interesting, the impression arose that the researchers have been in a great hurry to make this article and the final finalization has not been completed. This is highlighted e.g. in that in the text some of the references are numbered and some are indicated by the name of the author. 

There are also other issues:

-       Title needs to include the fact that this is re-analysis of four earlier carried out studies

-       Introduction is far too long. Try to occupy majority of the text in the discussion part.

-       COVID-19 issue in the introduction is not appropriate. Move it to discussion and shortan because it is not related to materials in this study.

-       I would like to see separate table showing with reference number description of the four included studies. In the current version there is only clear description of variables of these studies.

-       I understand that authors need to have ethical permission if they re-analyze data of these studies. If so state that this permission was available. If not, explain in the methods how this re-analysis can be carried without this kind of permission

-       I think that Table 2 represents already results but why it is in the methods?

-       I wish that authors clarify with figures the results. Currently results are not at all reader friendly form. F. ex. I try to understand Figure 1 and had difficulties. So, please, have clear figure/Table legends. The journal is only online so you are free to use colors too.

-       Limitations needs to include the fact that data has been gathered from four different studies

-       Make final conclusions more practical and clearly state in detail what needs to be studies next.

Reviewer 4 Report

Overall, this work is quite good. Only some issues are needed to be addressed.

  1. A lot of information about the impacts of COVID-19 on psychological distress, work-family conflict and supervisors supportive behaviours were mention in the introduction. Whether the data were collected during the outbreak of COVID-19 and the participants’ working lives were affected by the pandemic. If yes, COVID-19 should be mentioned in the title.
  2. What is the representativeness of these selected population? Why selecting these population as the target?
  3. What is the data collection process? What methods did you use to select these participants?
  4. This study is definitely able to provide the practical implications based on the findings. However, this part had not been discussed in detail. How can the relevant organisations, employers and supervisor do to improve workers’ psychological wellbeing regarding your findings?
  5. The conclusion is too brief. Please improve it.
